# Clinical Studies Using Topical Melatonin

**DOI:** 10.3390/ijms25105167

**Published:** 2024-05-09

**Authors:** Giovanni Greco, Ritamaria Di Lorenzo, Lucia Ricci, Teresa Di Serio, Eleonora Vardaro, Sonia Laneri

**Affiliations:** Dipartimento di Farmacia, Università degli Studi di Napoli Federico II, Via Domenico Montesano, 49, 80131 Napoli, Italy; ggreco@unina.it (G.G.); ritamaria.dilorenzo@unina.it (R.D.L.); lucia.ricci@unina.it (L.R.); teresa.diserio@unina.it (T.D.S.); eleonora.vardaro@unina.it (E.V.)

**Keywords:** melatonin, topical applications, clinical studies, skin, dermatology, photoprotection, anti-aging, hair growth

## Abstract

Melatonin is ubiquitously present in all animals and plants, where it exerts a variety of physiological activities thanks to its antioxidant properties and its key role as the first messenger of extracellular signaling functions. Most of the clinical studies on melatonin refer to its widespread oral use as a dietary supplement to improve sleep. A far smaller number of articles describe the clinical applications of topical melatonin to treat or prevent skin disorders by exploiting its antioxidant and anti-inflammatory activities. This review focuses on the clinical studies in which melatonin was applied on the skin as a photoprotective, anti-aging, or hair growth-promoting agent. The methodologies and results of such studies are discussed to provide an overall picture of the state of the art in this intriguing field of research. The clinical studies in which melatonin was applied on the skin before exposure to radiation (UV, sunlight, and high-energy beams) were all characterized by an appropriate design (randomized, double-blind, and placebo-controlled) and strongly support its clinical efficacy in preventing or reducing skin damage such as dermatitis, erythema, and sunburn. Most of the studies examined in this review do not provide a clear demonstration of the efficacy of topical melatonin as a skin anti-aging or as a hair growth-promoting agent owing to limitations in their design and/or to the use of melatonin combined with extra active ingredients, except for one trial that suggests a possible beneficial role of melatonin in treating some forms of alopecia in women. Further research efforts are required to reach definitive conclusions concerning the actual benefits of topical melatonin to counteract skin aging and hair loss.

## 1. Introduction

Melatonin is an endogenous transmitter, present in almost all organs and tissues, that regulates many physiological functions: (i) circadian sleep–wake and neuroendocrine cycles, (ii) blood pressure and autonomic cardiovascular activity, (iii) bone formation and protection, (iv) skin functions, (v) inflammatory response, and (vi) protection against oxidative stress [1]. Such a peculiar pharmacological profile results from a variety of mechanisms of action taking place in the various cell compartments, including the cytoplasm, mitochondria, and nucleus, as well as in the extracellular matrix. Melatonin (Figure 1) passively diffuses through the biological membranes thanks to its low molecular weight (232.3 Da), lipophilicity (logP_n-oct/w_ = 1.32), and lack of acid/base equilibrium ionization [2].

Most of the clinical studies on melatonin published to date refer to its widespread oral use as a dietary supplement to treat sleep disorders [3,4,5,6]. The therapeutic potential of oral melatonin has also been explored in a variety of clinical trials to treat diseases not related to sleep disorders, but none of them has ever led to approval by health authorities [6]. The use of topical melatonin has been evaluated in a far smaller number of clinical trials to treat or prevent skin disorders prompted by the evidence that human skin is characterized by high concentrations of melatonin acting as an agonist at specific G protein-coupled receptors [7]. Moreover, melatonin easily penetrates the skin [8] and, being an endogenous substance, is well tolerated and devoid of allergenic potential [4]. This review highlights the results obtained from clinical studies in which melatonin was applied on the skin as a photoprotective, anti-aging, or hair growth-promoting agent.

## 2. Methods

In this review, a total of 18 clinical trials were examined following a comprehensive electronic search carried out in December 2023 through the PubMed database employing the following query: (melatonin) AND (topical OR skin). No restrictions were imposed regarding the year of publication. The results were filtered with “clinical trial” selected as the “article type”. We retained papers describing the application of melatonin on human skin and discarded papers focusing on topics such as studies in animal models, studies evaluating the percutaneous penetration of melatonin, effects on melatonin serum levels, effects of melatonin on the skin when administered only orally, ex-vivo studies, and the application of melatonin on mucosae. This approach allowed us to retrieve 11 trials dealing with three main fields of applications of topical melatonin: photoprotection, skin aging, and alopecia. The PubMed data were further searched by using the following queries: (i) (melatonin) AND (photoprotection OR erythema OR dermatitis), (ii) (melatonin) AND (skin aging OR anti-aging), (iii) (melatonin) AND (alopecia OR hair loss OR hair growth) while keeping the “clinical trial” filter, thus obtaining an additional three trials related to the selected three fields of application of topical melatonin. Finally, we looked for potentially relevant references provided by all the retrieved articles, thus identifying four additional relevant trials.

## 3. Pharmacology of Melatonin

The complex pharmacology of melatonin has been thoroughly described in several excellent reviews, such as those by Tordjman et al. [1], Dubocovich et al. [9], and Cecon et al. [10]. Herein, we summarize the main mechanisms of action of melatonin underlying its antioxidant and anti-inflammatory properties, most of which are schematized in Figure 2.

Melatonin acts as an agonist at two G protein-coupled receptors called MT1 and MT2 and passively diffuses in the cell where it exerts of variety of effects. The activation of MT1 receptors decreases cAMP formation, protein kinase A (PKA) activity, and the phosphorylation of the cAMP responsive element-binding protein (CREB); the activation of MT2 receptors decreases cAMP and cGMP formation and increases protein kinase C (PKC) activity and the phosphorylation of the mitogen-activated protein kinases 1 and 2 (MAPK1 and MAPK2) [1,9,10].

Melatonin stimulates the expression of antioxidant enzymes, such as superoxide dismutase, catalase, and glutathione peroxidase [11,12]. The overexpression of these enzymes is related to the upregulation of the nuclear factor erythroid 2-related factor 2 (Nrf2) through the pathways mediated by the MT1 and MT2 receptors via the activation of PKC and calcium influx [13]. However, the upregulation of Nrf2 by melatonin also occurs independently from such receptors as it has been proved that Nrf2 increases when the MT1 and MT2 receptors are blocked by their antagonist luzindole [14]. Specifically, after melatonin and its metabolites have entered the cell, they upregulate Nrf2 by occupying the aryl hydrocarbon receptor (AHR) [15] and the peroxisome proliferator-activated receptor γ (PPARγ) [16] by acting as agonists at both receptors [17]. The role of these two receptors in protecting cells against metabolic and oxidative injuries has been clearly established [18]. The upregulation of Nrf2 by melatonin is also related to its capability to inhibit the ubiquitin–proteasome system (UPS) [19].

Melatonin counteracts the apoptotic activity of tumor necrosis factor-α (TNF-α) through a MAPK-dependent signaling cascade following the activation of the MT1 and MT2 receptors, as well as through its direct antioxidant action [20].

Melatonin binds to quinone reductase 2 (QR2), a cytosolic and ubiquitously expressed flavoprotein enzyme (previously hypothesized as the low-affinity putative MT3 binding site [21]), which exerts antioxidant and anti-apoptotic functions [22].

The antioxidant and anti-inflammatory properties of melatonin rely not only on its receptor-mediated actions but also on its radical scavenging activity against the reactive oxygen species (ROS) that threaten the integrity of mitochondria [23,24]. Melatonin reduces the release of apoptotic and pro-inflammatory interleukins by interfering with the NLPR2 inflammasome by modulating several transcription factors and neutralizing the ROS either directly as a radical scavenger or indirectly by upregulating Nrf2 [25,26].

The anti-inflammatory properties of melatonin are also related to its effects on the metabolism of arachidonic acid. Specifically, melatonin reduces the expression of cyclooxygenase-2 (COX-2) and inducible nitric oxide synthase (iNOS) by inhibiting the activity of p300 histone acetyltransferase (p300 HAT), thereby suppressing the acetylation of p52 and binding to the NF-κB transcription factor [27]. Finally, melatonin reduces the catalytic activity of metalloproteinase-9 by occupying its binding site [28].

Melatonin biosynthesis occurs not only in the pineal gland (from which it is secreted to regulate the circadian rhythm in mammals) but also in other organs like the retina, cochlea, lungs, liver, kidney, immune system, and skin [29]. Its plasmatic and tissue levels are finely tuned by metabolic reactions taking place mainly in the liver, kidney, and epidermis [1,30]. The elimination half-life of melatonin following oral or intravenous administration is less than one hour [31]; its oral bioavailability—referred to as a 2–4 mg dosage—is approximately 15% [32].

## 4. Rationale of Using Topical Melatonin to Treat Skin Conditions

Human skin is characterized by significant levels of melatonin (and are higher in African Americans than in Caucasians) and an efficient melatoninergic system [7,33]. The MT1 and MT2 receptors are expressed in the dermal fibroblasts, epidermal keratinocytes, hair follicles (the dermal papilla and the inner and outer root sheaths), eccrine glands, and blood vessels [7,34,35,36,37,38]. The skin is exposed to various external insults mainly consisting of solar radiation and environmental pollutants, which cause oxidative stress and inflammation, thus accelerating the photoaging processes [39,40,41,42,43]. Melatonin can easily penetrate the skin [8], where it can display its antioxidant and anti-inflammatory activities. Being an endogenous substance, melatonin is substantially safe and practically devoid of allergenic potential [4].

Topical melatonin has been investigated in clinical studies for its efficacy in protecting the skin from radiation, reducing skin aging, and promoting hair growth.

## 5. Melatonin as a Photoprotective and Anti-Inflammatory Agent

The antioxidant and anti-inflammatory properties of melatonin led several research groups to investigate its topical use as a photoprotective agent (see the clinical studies in Table 1).

The first report on the capability of melatonin to protect the skin from radiation was published by Bangha and coworkers [44]. Their randomized, double-blind, placebo-controlled study involved 20 volunteers with Fitzpatrick skin types II or III [45] who were exposed to UVB radiation on four areas in their lower back. Immediately after irradiation, the areas were treated with gels containing melatonin at different concentrations to reach 0.012 mg, 0.024 mg, and 0.12 mg of melatonin per cm^2^ of treated area. The vehicle alone was used as the placebo.

The UV-induced erythemas were assessed 8 and 24 h after exposure by a visual inspection using a modified score-based scale [46] and chromametry (the redness was expressed as a* value) [47]. The volunteers were divided into two groups of 10 according to the degree of the erythematous reaction in the placebo-treated area: high responders and low responders. An assessment of the visual and instrumental outcomes was performed separately in each of the two groups.

Appreciable results were obtained only in the group of high responders 8 h after irradiation, thus suggesting that topical melatonin (with the best concentration being 0.5%) may have a role as a skin photoprotective agent.

In a later article, Bangha et al. described how the time of the topical application is a key determinant of the photoprotective effect of melatonin [48]. This randomized, double-blind, placebo-controlled study was carried out on the same group of volunteers enrolled in the previous work [44]. Defined areas of their back were treated with 0.6 mg melatonin/cm^2^ or the vehicle alone. The gel was applied 15 min before or 1, 30, and 240 min after UV irradiation. Visual scoring and chromametry were used again to evaluate the erythemas 24 h after irradiation.

The melatonin applied 15 min before the UV irradiation almost completely suppressed the development of UV-induced erythemas in all the subjects, whereas no significant protective effects were observed when the same treatment was performed after UV irradiation. These results apparently contradict the findings of the previous study of the same authors [44] in which melatonin—applied immediately after exposure to UV radiation—was effective in a group of subjects characterized by a stronger skin reaction to UV radiation. In this regard, the authors outlined that the reason for the contrasting results could depend on different experimental conditions adopted in the two studies (different UV sources, time points of the application, and the amount of melatonin per treated area). However, the two studies under comparison become convergent in their results without splitting the volunteers into two groups (low and high responders) as performed in the previous work: taken together, the data of both studies show that melatonin applied in all the treated subjects was much more effective when applied before exposure to UV rays.

The superior performance of melatonin as a photoprotective agent when applied before irradiation likely stems from the possibility of this substance of efficiently neutralizing the ROS and metabolites of arachidonic acids just when their levels start to rise, rather than when they have already highly increased and the major skin damage has occurred.

Based on the results described in the articles by Bangha et al. [44,48], Dreher and coworkers conducted a randomized double-blind, placebo-controlled study to evaluate the efficacy of melatonin in preventing UV-induced erythemas when applied 30 min before irradiation [49]. A novelty in this study was that vitamin C and vitamin E were tested alone or in combination with the melatonin, seeking the potential synergistic effects among the three antioxidant ingredients. Twelve subjects with Fitzpatrick skin types II or III were enrolled. Several outcomes were employed to assess the skin response: chromametry, visual scoring, variation of the dermal blood flow, transepidermal water loss (TEWL), and electrical capacitance.

Differently from the above two reviewed studies by Bangha et al. [44,48], in this work the mid-backs of the volunteers were irradiated with an exact amount of energy defined as the minimal erythema dose of UV radiation that would determine a visual erythema score of 2. Such a procedure led to a homogeneous response (moderate redness with a sharp boundary) in the skin of all the subjects.

The active ingredients were dissolved in various concentrations (1% and 2.5% of melatonin, 2% of vitamin E, and 5% of vitamin C) in a hydroalcoholic vehicle. The vehicle alone was used as the placebo. Compared with the studies by Bangha et al. [44,48], in this work the amounts of melatonin per cm^2^ were much higher (2 or 5 mg versus 0.12 or 0.6 mg).

The three outcomes showed the same trend regarding the erythemal response 24 h after irradiation. Particularly, 2.5% of melatonin significantly protected the skin from UV radiation as the treated areas exhibited a mean redness a* value of 9.5 compared with 11.5 on the placebo control area (−17.4%). These data confirmed the finding of Bangha et al. [48] regarding the use of melatonin as a photoprotective agent when applied before exposure to radiation.

The formulation containing 2.5% of melatonin, 2% of vitamin E, and 5% of vitamin C showed the greatest efficacy. Specifically, such a combination led to a mean redness a* value 32.2% lower than the placebo-treated area. The pronounced effect obtained by combining melatonin with vitamin E and vitamin C was rationalized by the authors considering that vitamin E is a well-known photoprotective agent while vitamin C regenerates the radical scavenging activity of vitamin E.

No significant variations in the skin TEWL and electrical capacitance with respect to the controls were observed for all the antioxidant-treated areas.

Melatonin showed UV absorption in a dose-dependent manner. Specifically, the 1% and 2.5% melatonin formulations exhibited sun protection factors of 1.47 and 2.1, respectively. These data demonstrate that the sunscreen property of melatonin contributes moderately to protect the skin from UV radiation.

In a subsequent study, Dreher et al. tested the photoprotective effects of melatonin, vitamin E, and vitamin C when applied after exposure to UV radiation [50]. In this randomized, double-blind, placebo-controlled trial, the researchers employed similar experimental conditions to their previous study [49], except for the number of volunteers (six subjects with Fitzpatrick skin types II or III). No significant differences were observed between the antioxidant- and the placebo-treated arms. These results are consistent with the former work by Bangha et al. [44] who did not find statistically significant photoprotective effects exerted by melatonin in the whole panel of volunteers (either low and high responders) when applied after exposition to UV rays.

The authors stated that the UV-induced erythema, triggered by a massive generation of ROS, is such a rapid process that can be efficiently prevented when the antioxidants are already present in the skin before the UV irradiation.

The need to apply melatonin before exposure to radiation was confirmed by a randomized, double-blind, placebo-controlled study by Howes et al. [51], who found that melatonin (0.1 mg/cm^2^) failed to protect the skin from UV rays when applied immediately after irradiation.

Scheuer et al. [52] investigated the efficacy of topical melatonin in preventing solar erythema. Three different concentrations of a melatonin cream (0.5%, 2.5%, and 12.5%) were used to investigate the dose–response relationship of the melatonin used to prevent skin damage caused by radiation. This randomized, double-blind, placebo-controlled study involved 21 volunteers (those who completed the study) with Fitzpatrick skin types I, II, or III. The subjects had their back exposed to the sun for 40 min when the UV index was 9. All the participants applied the melatonin preparations or the vehicle alone on specific areas of their back before exposure to the sunlight. The amounts of melatonin per cm^2^ of skin were 1.3 mg, 6.64 mg, and 33.20 mg. The cream was devoid of substances with sunscreen properties.

The participants were divided into two groups depending on the erythema reaction following sun exposure as assessed by chromametry: 11 low responders (changes in the a* value < 5) and 10 high responders (changes in the a* value > 5).

The maximal erythemal reaction was found for all the participants 8 h after sun exposure. The outcomes for all the 21 subjects showed no significant differences. Instead, in the group of high responders the areas treated with the 12.5% melatonin cream showed significantly fewer erythemas compared with the vehicle-treated and untreated areas at all time points. No beneficial effects were observed using 0.5% or 2.5% of melatonin.

The conclusion of this work might sound like it conflicts with that of Dreher et al. [49] using UV light, which showed that melatonin was effective in all the treated volunteers without any splitting of the volunteers into low and high responders. However, there are two factors to be considered that can explain such a difference in the results of the two studies. First, the volunteers enrolled by Scheuer had Fitzpatrick types I, II, or III, whereas in the study by Dreher the volunteers were of Fitzpatrick types II and III. Furthermore, the type of irradiation was different in the two trials: sunlight in the former and a dose of UV radiation determining a predefined inflammatory response in the latter. Both factors may have contributed to more heterogeneous responses to sunlight in the study by Scheuer, such as to limit the statistical significance of the results of the whole panel of 21 subjects.

The safety profile of the cream containing 12.5% of melatonin was subsequently assessed by Scheuer’s group in a randomized, double-blind, placebo-controlled, crossover study conducted on 10 volunteers [53]. The authors investigated whether cognitive functions could be affected by large amounts of melatonin absorbed percutaneously. Each participant was randomized to apply the cream containing melatonin or the vehicle alone on 80% of the body surface. The crossover setting was carried out by taking an interval of 2 weeks between the application of melatonin or the placebo.

The melatonin doses ranged between 4.1 g and 5.6 g depending on the height and weight of each subject, thus leading to the plasma concentrations of melatonin—sampled 12 h after the topical administration—reaching a mean value of 5 ng/mL. Such a level is much higher than the physiological levels of melatonin ranging from 10 pg/mL in daylight hours to 50–70 pg/mL in the night [54].

The degree of cognitive functions was assessed during daylight hours using three outcomes: the Karolinska sleepiness scale [55], finger tapping test [56], and continuous reaction time [57]. The volunteers were also monitored to identify possible adverse reactions such as headache, tiredness, dizziness, and confusion. No significant cognitive dysfunctions or side effects were detected in the treatment with melatonin compared with the placebo.

The lack of sedative effects upon absorption of huge doses of melatonin was discussed by the authors. They hypothesized that a greater sedation is caused by melatonin when it is taken in the evening or before going to bed rather than in the early morning [58] as was performed in their setting. Additionally, the sedative effects of melatonin may be restricted to a narrow range of low doses or, alternatively, there may be a threshold level of plasmatic melatonin beyond which the dose–response curve is substantially flat.

Cancer radiotherapy consists in irradiating specific areas of the body to kill neoplastic cells with high-energy X-ray rays, gamma rays, or charged particles (electrons or protons). The main side effect of radiotherapy is dermatitis, whose severity is graded in a continuum, ranging from erythema and dry desquamation to more severe desquamation and, eventually, ulceration [59,60]. This condition is characterized by symptoms such as skin dryness, itching, discomfort, pain, warmth, and burning, which may lead to the cessation of the treatment and may persist up to a month after its completion [61]. Two articles describe the use of topical melatonin to minimize dermatitis caused by radiotherapy in women affected by breast cancer following curative surgery. A limitation in both the quoted studies is represented by a clinical assessment of the outcomes not supported by instrumental measurements.

Ben-David et al. conducted a randomized, double-blind, placebo-controlled study to evaluate the effects of melatonin in 47 women undergoing radiotherapy, each receiving a total of 50 Gy of radiation [62]. The volunteers applied a melatonin emulsion (26 subjects) or placebo (21 subjects) on their breast twice daily during the radiation treatment and 2 weeks following the end of the radiotherapy.

The occurrence of grade 1–2 acute dermatitis was significantly lower (59% vs. 90%) in the melatonin group. A drawback of this study is that the melatonin dosage was not specified.

Recently, Zetner et al. reported the results of a randomized, double-blind, placebo-controlled trial in which melatonin was employed to reduce the side effects caused by radiotherapy in subjects receiving a total of either 40.5 Gy or 50 Gy of radiation [63]. There were 65 women who were enrolled in this study but 17 dropped out, thus leading to 26 and 22 patients in the melatonin and placebo groups, respectively. They applied 1 g of cream containing melatonin or the vehicle alone on the irradiated area of the skin at least 2 h before irradiation. The cream was used every day from the beginning of the radiotherapy till 2 weeks after its conclusion. The melatonin cream contained 2.5% of melatonin and 1.5% of dimethyl sulfoxide. A breast symptom (BS) score (proportional to the severity of the radiodermatitis) was clinically evaluated at weeks 1, 2, 3, 4, and 5; the last radiation fraction; and follow-up weeks 1, 2, and 3, with week 1 serving as the baseline level. The primary outcome was the BS score measured on the day of the last radiation fraction. The BS scores were also analyzed over time throughout the entire period of radiotherapy, including 3 weeks of follow-up, as a secondary outcome.

In the melatonin-treated patients, the mean BS score was always at least 6 points lower than in the placebo-treated patients from week 2 to week 4 and during the entire follow-up period, except for week 5 (the last radiation fraction) when this distance dropped to 4 points. The differences in the mean BS scores in the two arms were not sufficient to confer statistical significance to the primary outcome but turned out to be as significant for the secondary outcome. In other words, the patients using melatonin experienced fewer side effects related to skin damage during most of the study period. Incidentally, dimethyl sulfoxide is endowed with antioxidant properties [64] and might have contributed to a certain extent to the beneficial effects of the cream containing melatonin.

**Table 1 ijms-25-05167-t001:** Clinical trials reporting the photoprotective and anti-inflammatory efficacy assessment of melatonin for topical treatment (all randomized, double-blind, and placebo-controlled).

Topical Interventionand Dosage ^a^	SkinCondition	Measured Parameters	Differences from the Control (*p* Values) ^b^	Reference
0.5% melatonin (0.12 mg/cm^2^)after irradiation	UV-induced erythema	all volunteers		Bangha et al., 1996 [44]
erythema (SBCA) ^c^	ns
erythema a*—redness	ns
high responders	
erythema (SBCA) ^c^	*p* < 0.05
erythema a*—redness	*p* < 0.05
0.5% melatonin (0.6 mg/cm^2^)before irradiation	UV-induced erythema	erythema (SBCA) ^c^	*p* < 0.01*p* < 0.01*p* < 0.01	Bangha et al., 1997 [48]
erythema a*—redness
erythema L*—luminance
2.5% melatonin (5 mg/cm^2^) 2% vitamin E5% vitamin Cbefore irradiation	UV-induced erythema	erythema (SBCA) ^c^	*p* < 0.05	Dreher et al., 1998 [49]
dermal blood flow	*p* < 0.05
erythema a*—redness	*p* < 0.05
transepidermal water loss	ns
electrical capacitance	ns
2.5% melatonin (5 mg/cm^2^)2% vitamin E5% vitamin Cafter irradiation	UV-induced erythema	erythema (SBCA) ^c^erythema a*—rednessdermal blood flow	nsnsns	Dreher et al., 1999 [50]
5% melatonin (0.1 mg/cm^2^)after irradiation	UV-induced erythema	erythema indexerythema (SBCA) ^c^	nsns	Howes et al., 2006 [51]
12.5% melatonin (33.2 mg/cm^2^)before exposure	solar erythema	all volunteers		Scheuer et al., 2016 [52]

erythema a*—redness	ns
erythema (SBCA) ^c^	ns
high responderserythema a*—rednesserythema (SBCA) ^c^	
*p* = 0.013*p* = 0.02
melatonin (ng)before irradiation	radiodermatitis	dermatitis (SBCA) ^c^	*p* < 0.05	Ben-David et al., 2016 [62]
2.5% melatonin1.5% dimethyl sulfoxidebefore irradiation	radiodermatitis	breast symptom score (SBCA) ^c^	*p* < 0.5	Zetner et al., 2023 [63]

^a^ Dosage yielding best results; ng: not given concentration. ^b^ According to the results and conclusions reported by authors; ns: not significant. ^c^ Score-based clinical assessment.

Overall, the conclusion of this study concerning the clinical efficacy of melatonin in minimizing the side effects of radiation therapy may be considered consistent with the results of the previously quoted study by Ben-David et al. [62].

Based on the clinical studies reviewed in this section, characterized by sound design (randomized, double-blind, and placebo-controlled), it can be safely said that melatonin applied on the skin before exposure to radiation (UV, sunlight, and high-energy beams) is effective in preventing or reducing damage such as dermatitis, erythema, and sunburn. The only weak point of the studies evaluating the photoprotective efficacy of melatonin against UV or sunlight is the relatively low numbers of volunteers being examined (no more than 21). Nevertheless, such studies were all based on a more robust evaluation of the outcomes compared with those testing melatonin against high-energy beams used in radiotherapy, because the former ones were evaluated not only through clinical assessment but also by instrumental measurements (i.e., chromametry).

Melatonin concentrations from to 0.5% to 12.5% have been tested and a clear direct dose–response relationship has been found. Statistically significant beneficial effects have been observed using amounts of 0.6 mg up to 33.2 mg per cm^2^ of treated skin. The addition of vitamins D and C to melatonin-containing formulations leads to a synergistic improvement of the photoprotective effects.

Melatonin is absorbed percutaneously. However, even high doses of melatonin applied on the skin (up to 5.6 g) do not cause local or systemic side effects. 

The photoprotective action of melatonin can be explained in terms of its capability to neutralize the massive release of ROS upon exposure to UV radiation and to minimize ROS-mediated inflammation responses thanks to its antioxidant and anti-inflammatory properties.

## 6. Melatonin as a Skin Anti-Aging Agent

The experimental evidence that melatonin represents a valid photoprotective agent led to investigations aimed at exploring its skin anti-aging potential as skin aging is largely dependent on exposure to sunlight. More specifically, the UV components of solar radiation have been reported to contribute by about 80% to facial aging [65]. Solar rays, like other factors such as air pollution, the use of harsh cosmetics, toxic substances produced during smoking, and alcohol consumption, cause skin damage resulting from chronically sustained levels of free radicals [66]. These free radicals, in turn, trigger inflammatory responses that affect the metabolism of collagen and elastin, thus leading the skin to lose flexibility and strength, as well as to gain roughness, dryness, irregular pigmentation, and deep wrinkling [67]. With respect to these phenomena, melatonin was expected by several researchers to play a beneficial role as a skin anti-aging agent owing to its ascertained antioxidant and anti-inflammatory activities together with an excellent safety profile for its long-term topical use.

The clinical trials discussed below—aimed at evaluating melatonin for its skin anti-aging efficacy—are summarized in Table 2.

Sagan et al. investigated topical melatonin for its ability to reverse skin oxidative stress [68] in an open-label trial. The treatments consisted of topical melatonin (a 0.012% aqueous solution), oral melatonin (2.5 mg before bedtime), and a commercially available product (Restructurer) containing the following antioxidant and healing agents: 3% of ascorbic acid, 0.5% of methylsilanol mannuonate, 0.173% of zinc gluconate, 0.01% of thioctic acid, and 0.01% of manganese gluconate.

Ninety women were enrolled and divided into two groups: former smokers and never-smokers. Such a split allowed the researchers to investigate the effects of the treatments on the skin affected or not affected by oxidative stress related to cigarette smoking, respectively. Each of the two major groups of volunteers was further divided into four subgroups: (i) control (no treatment), (ii) topical melatonin, (iii) oral melatonin, and (iv) Restructurer.

Microdermabrasions were performed at three time points: point 0 (the baseline), after 2 weeks, and after 4 weeks. At the same time points, the following parameters were measured: lipid peroxidation (LPO) in the blood serum, LPO in the epidermis exfoliated during the microdermabrasion, and four skin biophysical characteristics (measured instrumentally), such as the sebum, moisture, elasticity, and pigmentation. LPO was considered as a marker of oxidative damage. Topical melatonin (5 mL) and Restructurer (5 mL) were applied on the skin just after the microdermabrasion.

The four types of treatments revealed statistically significant differences in the outcomes—compared with the control group—only in the major group of former smokers. Particularly, oral melatonin was effective in decreasing the serum LPO and improving the sebum, moisture, and elasticity but did not influence the LPO in the epidermis and pigmentation. Topical melatonin only improved the sebum levels, thus showing itself to be less performant than oral melatonin in ameliorating the skin biophysical parameters. The Restructurer only increased the elasticity.

As admitted by the authors, the small sample size (eight to fourteen subjects in the subgroup of former smokers) represents a limitation to the validity of their conclusions. This study suggests that a short period (4 weeks) of treatment with topical melatonin does not protect the skin from the oxidative stress caused by cigarette smoking. However, such a finding might depend not only on the short period of the study but also on the use of a melatonin concentration (0.012%) much lower that the concentrations (0.5–12.5%) employed in the trials that proved its efficacy as a photoprotective agent (see previous section). The issue of the concentration of the melatonin employed in this study is particularly relevant to reach definite conclusions about its usefulness as a topical skin anti-aging agent considering that in the same study oral melatonin was indeed demonstrated to be effective in improving all the skin biophysical characteristics being examined.

Differently from the work by Sagan et al. [68], the four trials described below used formulations containing melatonin combined with additional active components (mainly antioxidant substances).

Morganti et al. conducted a randomized, double-blind, placebo-controlled study to evaluate the anti-aging effects of melatonin, vitamin E, and β-glucan [69]. This last ingredient has been reported as an immunomodulator [70,71] promoting skin health [72]. The three active components were mixed with hyaluronic acid or chitin to obtain two types of emulsions and two types of capsules. The final concentration/amount of each active component was 0.0002% or 1.6 mg, respectively. Seventy women were enrolled and randomly divided into seven arms to receive twice daily active/placebo emulsions and/or active/placebo capsules containing or lacking chitin.

After 12 weeks of treatment, the topical formulations led to statistically significant improvements of the skin biophysical parameters and photoaging appearance compared with the placebo group. The topical treatments showed better results compared with the oral treatments, and all the formulations containing chitin performed better than those mixed with hyaluronic acid. The best results were obtained with the topical plus oral treatments.

The results of this study are surprisingly remarkable given the extremely low concentration of topical melatonin being employed (0.0002%) compared with the concentrations adopted in all the other clinical studies examined in this review (ranging from 0.0033% to 12.5%). As mentioned, the topical treatments in this trial turned out superior to oral ones (3.2 mg/daily) in improving the biophysical parameters of the skin. Notice that in the previously reviewed study by Sagan et al. [68] a considerably higher concentration of topical melatonin (0.012%, without any additional active ingredient) was found less effective than oral melatonin (2.5 mg/daily). These contrasting results might only be partially explained in terms of the synergistic effects resulting from the combination of melatonin with vitamin E and β-glucan employed in the study by Morganti. If this were the case, it should be assumed that in the study by Morganti the same combination of active ingredients administered orally reached the skin in lower concentrations than those obtained by topical application. However, such a hypothesis would conflict with that observed in the study by Sagan.

Two commercially available products containing 0.1% of melatonin were investigated for their skin anti-aging effects by Milani and coworkers [73]. Melatonin was incorporated in lipospheres made of vegetal oils endowed with antioxidant properties. This randomized, split-face, open-label study was conducted on 22 women with moderate–severe skin aging. The two products were applied once in the morning (day cream) and once in the evening (night cream) on the right or the left side of the face according to a randomization procedure.

Several skin parameters were assessed by comparing the treated with the untreated face after 3 months: (i) dryness, (ii) firmness/tonicity, (iii) surface microrelief, (iv) a computer-aided evaluation of wrinkles (crow’s feet and nasolabial folds), and (v) roughness. The first four outcomes were evaluated clinically, whereas the last one instrumentally. Statistically significant improvements were observed in the first four outcomes resulting from the treatment compared with no treatment.

Goldberg et al. evaluated the anti-aging efficacy of a serum-in-oil emulsion containing melatonin, a bakuchiol extract, and ascorbyl tetraidroisopalmitate [74] in an open-label trial. Bakuchiol is a meroterpene phenol obtained from the seeds of *Psoralea corylifolia* endowed with antioxidant, anti-inflammatory, and antibacterial activities [75], which proved to be equipotent with retinol in the treatment of facial photoaging [76]. Ascorbyl tetraisopalmitate is a lipophilic prodrug of vitamin C, which penetrates the skin and improves hydration and microrelief [77]. The concentrations of these three active ingredients incorporated in the emulsion were not given. In this open-label study, 39 women with Fitzpatrick skin types I–III applied 4–5 drops of emulsion on their face each evening for 12 weeks. They were also instructed to use a sunscreen product with a sun protection factor of at least 50 when exposed to sunlight.

Three outcomes were assessed after 4, 8, and 12 weeks to be compared with the pretreatment baseline: (i) wrinkles in the crow’s feet area, (ii) pigmentation, and (iii) firmness. The first two were measured instrumentally, whereas the last one was evaluated according to a score-based scale. At the three time points, all the outcomes revealed that the serum-in-oil emulsion produced significant beneficial effects.

Granger et al. tested an oil-in-water emulsion night cream containing melatonin and additional active ingredients for its beneficial effects on skin [78]. The additional components of the cream were niacinamide, hyaluronic acid, carnosine, matricin peptides, and an extract of *Helichrysum italicum*. The methodologies of this clinical study were similar to those adopted by Goldberg et al. [74], except that no sunscreen product was part of the treatment. After 12 weeks of cream-based treatment, all the outcomes improved with respect to the pretreatment baseline.

The five clinical studies reviewed in this section evaluated topical melatonin as a skin anti-aging agent. Among them, only the study by Morganti et al. [69] was randomized, double-blind, and placebo-controlled, whereas the remaining ones [68,73,74,78] were open-label in their design and therefore of limited internal validity. The study by Sagan [68], the only one using topical melatonin as a single ingredient, found no significant benefits to the skin, except for an improvement in sebum levels compared with the untreated arm. In the remaining four studies [69,73,74,78], melatonin was applied on the skin combined with additional active components, thus hindering a precise estimation of the potential beneficial effects of melatonin alone. The studies by Goldberg et al. [74] and Granger et al. [78] suffered from two additional limitations: they were non-comparative trials in which a pretreatment baseline, rather than a placebo-treated or an untreated arm, was adopted as the control; moreover, the concentrations of melatonin and the additional active ingredients were not specified.

To sum up, none of the studies reported to date in the literature provides a clear demonstration of the efficacy of topical melatonin as a skin anti-aging agent. Further rigorously designed clinical trials should be undertaken to support such a claim on more robust experimental bases.

**Table 2 ijms-25-05167-t002:** Clinical trials reporting the anti-aging efficacy assessment of melatonin for topical treatment.

Topical Intervention and Dosage ^a^	Skin Condition	Trial Design ^b^	Measured Parameters	Differences from the Control (*p* Values) ^c^	Reference
0.012% melatonin	oxidative stress from cigarette smoking	OL, AC, NTC	lipid peroxidation	ns	Sagan et al., 2017 [68]
			sebum	*p* < 0.05	
			moisture	ns	
			elasticity	ns	
			pigmentation	ns	
0.0002% melatonin 0.0002% vitamin E 0.0002% b-glucan	skin aging	R, DB, PC	hydration	*p* < 0.005	Morganti et al., 2012 [69]
			lipids	*p* < 0.005	
			elasticity	*p* < 0.005	
			lipid peroxidation	*p* < 0.005	
			fine wrinkles (SBCA) ^d^	*p* < 0.005	
			senile dryness (SBCA) ^d^	*p* < 0.005	
			skin atrophy (SBCA) ^d^	*p* < 0.005	
			black spots (SBCA) ^d^	*p* < 0.005	
			telangiectasia (SBCA) ^d^	*p* < 0.005	
0.1% melatonin in lipospheres	skin aging	R, OL, SF, AB	wrinkles	*p* = 0.05	Milani et al., 2018 [73]
			microrelief	*p* = 0.001	
			tonicity (SBCA) ^d^	*p* = 0.05	
			dryness (SBCA) ^d^	*p* = 0.01	
melatonin (ng) ascorbyl tetraisopalmitate (ng) bakuchiol (ng)	skin aging	OL, NC	R_z_—average relief height	*p* = 0.09	Goldberg et al., 2019 [74]
			R_t_—maximum relief height	*p* = 0.03	
			deformation volume	*p* < 0.01	
			deformation depth	*p* < 0.01	
			L*—lightness	*p* < 0.01	
			b*—blue/yellow color	*p* < 0.01	
			individual typology angle	*p* < 0.01	
			pigmentation	*p* < 0.01	
			hydration	*p* < 0.05	
			transepidermal water loss	*p* < 0.05	
			sebum	*p* < 0.01	
			wrinkles (SBCA) ^d^	*p* < 0.01	
			firmness (SBCA) ^d^	*p* < 0.01	
			redness (SBCA) ^d^	*p* < 0.01	
melatonin (ng) carnosine (ng) *Helichrysum italicum* (ng)	skin aging	OL, NC	hydration	*p* < 0.05	Granger et al., 2020 [78]
			transepidermal water loss	ns	
			wrinkles count	*p* < 0.05	
			wrinkles volume	*p* < 0.05	
			wrinkles depth	*p* < 0.05	
			R_a_—arithmetic mean roughness	*p* < 0.05	
			R_z_—roughness depth	ns	
			brown spot count	*p* < 0.05	
			brown spot area	*p* < 0.05	
			UV spot count	*p* < 0.05	
			UV spot area	*p* < 0.05	
			R_0_—firmness	*p* < 0.05	
			R_2_—elasticity	*p* < 0.05	
			stinging score (SBCA) ^d^	*p* < 0.01	
			dryness (SBCA) ^d^	*p* < 0.01	
			erythema (SBCA) ^d^	*p* < 0.01	
			desquamation (SBCA) ^d^	*p* < 0.01	
			roughness (SBCA) ^d^	*p* < 0.01	

^a^ Dosage yielding best results; ng: not given. ^b^ AB: assessor blind; AC: active controlled; DB: double blind; NC: non-comparative; NTC: no treatment controlled; OL: open label; PC: placebo controlled; R: randomized; and SF: split face. ^c^ According to the results and conclusions reported by authors; ns: not significant. ^d^ Score-based clinical assessment.

## 7. Melatonin as a Topical Agent to Treat Alopecia

Androgenetic alopecia is a form of hair loss in both men and women associated with overactivity of dihydrotestosterone [79]. The treatment of androgenetic alopecia is usually based on two drugs approved by the FDA and EMA: oral finasteride (for men) and topical minoxidil (for men and women), which have a limited efficacy [80] and can cause side effects: gynecomastia in the case of finasteride [81] and irritation and an increased heart rate in the case of minoxidil [82,83]. Alopecia areata or diffuse alopecia is a chronic inflammatory disease characterized by non-scarring hair loss with preservation of the hair follicle seemingly related to autoimmune, genetic, and emotional stress, as well as endocrine factors [84,85]. There are currently no effective therapies for diffuse alopecia. 

Topical melatonin has been investigated as an alternative therapeutic agent to treat alopecia based on the following evidence. Melatonin regulates hair growth, cycle regulation, and pigmentation in animals [86,87,88]. The expression of MT1 and MT2 receptors is dependent on the hair cycle, and the melatonin levels in human hair follicles are higher than those in the serum, thus supporting a key role of melatonin in regulating hair growth [89]. In humans, the expression of melatonin receptors is higher in the dermal papilla than in the epidermal keratinocytes and dermal fibroblasts; their occupation activates hair growth-related genes [37]. Moreover, melatonin exerts anti-androgenic effects by facilitating the translocation of the androgen receptor from the nucleus to the cytoplasm and disrupting the positive interaction between the androgen receptor splice variant-7 expression and the activated nuclear factor-κB/interleukin-6 signaling [90,91,92].

The main features of the clinical studies discussed in this section are reported in Table 3.

Fischer and coworkers evaluated the efficacy of topical melatonin in reversing androgenetic and diffuse alopecia in women in a randomized, double-blind, placebo-controlled study [93]. They enrolled 40 women with androgenetic alopecia (12 subjects) or diffuse alopecia (28 subjects). These volunteers were divided into two groups of 20 depending on whether they were treated with melatonin or the placebo. An alcoholic solution containing 0.1% of melatonin or lacking this substance, as the placebo, was applied in the frontal and occipital areas of the scalp (1 mL, once daily in the evening). Trichograms were taken before the treatment and after 3 and 6 months. The anagen hair count compared to the non-anagen hair count, expressed as the odds ratio (OR), was the outcome adopted to evaluate the results of the study.

The maximum effects were observed after 6 months. The women with androgenetic alopecia applying melatonin on the occipital area exhibited a marked increase in anagen hairs (+8.7%) compared with a lower increase (+3.9%) given by the placebo. These differences yielded an OR value of 1.90. In the women with diffuse alopecia, melatonin increased the frontal hair counts by 1.6% compared with a 2.1% decrease resulting from the placebo treatment, corresponding to an OR value of 1.41. Taken together, these data indicate that melatonin can be more effective in treating androgenetic rather than diffuse alopecia in women. No significant effects were found in the frontal hair counts of women with androgenetic alopecia and in the occipital hair counts of those with diffuse alopecia.

The authors explained the lack of efficacy of the melatonin in the frontal area of the women suffering from androgenetic alopecia as due to the hypersensitivity of their hair roots to androgens resulting from an extremely high number of androgen receptors expressed by the dermal papilla cells of the frontal region of the scalp [94]. Regarding diffuse alopecia, melatonin was not effective in increasing hair growth in the frontal scalp for unknown reasons.

Melatonin applied on the scalp was absorbed percutaneously as its plasma levels were significantly higher in the melatonin-treated group (35–50 pg/mL) than in the placebo-treated group (5–10 pg/mL).

No information regarding the time of year during which the study was conducted was given, although this represents a relevant factor influencing human hair growth [95] as shown by the differences in hair counts within the placebo-treated groups. However, the double-blind, placebo-controlled setting could likely have minimized the influence of seasonal effects on the outcomes. A drawback with the trial setting was the low number of volunteers enrolled, such that further investigations would be desirable to confirm its results. However, this study provides a valuable contribution to our knowledge about the potential of melatonin as a hair growth promoter as it is the only one among those reported so far in the literature characterized by two key features: melatonin was used as a single ingredient and a sound design was adopted.

Four articles were subsequently published reporting the clinical evaluation of melatonin used alone or in combination with additional active ingredients to treat androgenetic alopecia. Unfortunately, all of them were non-comparative, open-label trials based on a pretreatment baseline as the control. Such a design represents a serious limitation to interpretation of the results owing to the unavoidable seasonal effects on hair growth that can be identified only by comparing a treated with an untreated arm.

Baldari et al. investigated the use of melatonin in the early stages of male androgenetic alopecia [96]. This open-label study enrolled 31 men suffering from androgenetic alopecia of types II or III of the Hamilton scale. The subjects applied 0.1 mg of melatonin, contained in a commercially available hydroalcoholic solution on the scalp every night for six months.

The hairs were counted on the parietal area in 16 patients and in the frontal area in 15 patients. The selected areas were those in which there was less hair density. In the parietal group, the mean hair density increased from 142 to 158 after 3 months and then fell slightly back to 153 after 6 months of treatment. A similar biphasic trend, although more pronounced, was also observed in the frontal group as the hair density was 153, 160, and 133 at the baseline and after 3 and 6 months of treatment. Notice that in this latter group the outcome markedly worsened upon completion of the treatment, thus suggesting a trend of hair growth likely related to seasonal effects. In both groups, the results were not statistically significant.

An article by Fischer et al. described the results of five clinical trials in which a 0.0033% melatonin solution was used to treat androgenetic alopecia in men and women [97]. The melatonin solution also contained biotin and a *Ginko biloba* extract (the concentrations of the last two active ingredients were not given). One of the trials was of particular interest as its outcomes consisted of instrumental measurements. In this open-label study, a group of 35 men with stage I or II androgenetic alopecia applied the hair solution on the scalp once daily for 6 months. At the end of this treatment, the hair count and density increased by 42.7% and 40.9%, respectively.

Hatem et al. investigated a formulation of nanostructured lipid carriers (NLCs) incorporating 2.5% of melatonin to treat mild to moderate male androgenetic alopecia [98]. A buffered 7.4 aqueous solution of melatonin—in the same concentrations as that incorporated in the NLC formulation—was likewise tested. This open-label study was conducted on 40 men. The subjects were randomly divided into two groups of 20. They applied the melatonin solution or the NLC formulation on the affected scalp once daily for 3 months. An assessment of the various outcomes (a hair pull test [99] and hystometric and dermoscopic examination) showed that both melatonin-based treatments increased the hair density and thickness and decreased the hair loss. Melatonin incorporated in the NLC formulation turned out to be more effective than the melatonin aqueous solution.

Hatem’s group also evaluated a topical formulation to treat androgenetic alopecia in which 10.6% of melatonin was incorporated into multilamellar nanovesicles containing ascorbyl palmitate (aspasomes) [100]. Forty men with stages I–V of alopecia were enrolled in this open-label study. The volunteers were equally divided into two groups to apply once daily the aspasomal formulation or an aqueous solution containing melatonin at the same concentration. At the end of 3 months of treatment, all the outcomes (the same as the previous work [98]) were significantly superior in both groups compared with the pretreatment baseline, with the aspasomal formulation performing better than the aqueous solution.

The above two reviewed works by Hatem et al. [98,100] suggest that advanced delivery formulations improve the penetration of melatonin into the skin compared with aqueous solutions. However, this finding leaves unsolved the question of whether melatonin (regardless of its formulation) might be effective in promoting hair growth owing to the non-comparative design of the quoted studies.

The literature offers a few articles describing the clinical evaluation of melatonin to promote hair growth. The study by Fischer et al. [93] (conducted on women with androgenetic alopecia and diffuse alopecia) is the only one in which melatonin was tested as a single ingredient and a sound design was adopted. The remaining four studies reviewed in this section [96,97,98,100] suffer from a weak design as they are non-comparative, open-label trials. Moreover, in one of them [97] the presence of active ingredients in addition to melatonin (unspecified concentrations of biotin and *Ginko biloba* extract) represents a confounding factor to estimate the potential beneficial effects of melatonin alone.

To sum up, there is to date little experimental evidence supporting the clear role of melatonin as a hair growth-promoting agent. Nevertheless, the above quoted article by Fischer et al. [93] suggests that melatonin may be beneficial in treating some forms of alopecia in women and that additional efforts aimed at evaluating the efficacy of melatonin in this field of application are worth undertaking.

**Table 3 ijms-25-05167-t003:** Clinical trials reporting the efficacy assessment of melatonin for the topical treatment of alopecia.

Topical Intervention and Dosage ^a^	Skin Condition	Trial Design ^b^	Measured Parameters	Differences from the Control (*p* Values) ^c^	Reference
0.1% melatonin	female androgenetic and diffuse alopecia	R, DB, PC	androgenetic alopecia		Fischer et al., 2004 [93]
			anagen/non-anagen hair count (frontal area)	ns	
			anagen/non-anagen hair count (occipital area)	*p* = 0.012	
			diffused alopecia		
			anagen/non-anagen hair count (frontal area)	*p* = 0.046	
			anagen/non-anagen hair count (occipital area)	ns	
0.1 mg melatonin	male androgenetic alopecia	OL, NC	hair density (parietal area)	ns	Baldari et al., 2007 [96]
			hair density (frontal area)	ns	
0.0033% melatonin biotin (ng) *Ginko biloba* (ng)	female and male androgenetic alopecia	OL, NC	hair density hair count	*p* < 0.01 *p* < 0.01	Fischer et al., 2012 [97]
2.5% melatonin in nanostructured lipid carriers	male androgenetic alopecia	OL, NC	baldness degree (SBCA) ^d^	*p* < 0.05	Hatem et al., 2018 [98]
			hair loss	*p* < 0.05	
			hair thickness	*p* < 0.05	
			sebaceus debris	*p* < 0.05	
			hair density	*p* < 0.05	
10.6% melatonin in ascorbyl palmitate multilamellar nanovesicles	male androgenetic alopecia	OL, NC	baldness degree (SBCA) ^d^	*p* < 0.05	Hatem et al., 2018 [100]
			hair loss	*p* < 0.05	
			hair thickness	*p* < 0.05	
			sebaceus debris	*p* < 0.05	
			hair density	*p* < 0.05	

^a^ Dosage yielding best results; ng: not given. ^b^ DB: double blind; NC: non-comparative; OL: open label; PC: placebo controlled; and R: randomized. ^c^ According to the results and conclusions reported by authors; ns: not significant. ^d^ Score-based clinical assessment.

## 8. Achievements, Limitations, and Future Prospects of Clinical Studies on Topical Melatonin

Satisfactory achievements in the clinical studies using topical melatonin were obtained when all or most of the following main experimental requirements were met: (i) sound design (i.e., randomized, double-blind, and placebo-controlled), (ii) adequate sample size, (iii) application of at least one formulation of melatonin as a single active ingredient, and (iv) instrumental measurement of the outcome/s. Such conditions were mostly fulfilled in studies in which melatonin was tested as a photoprotective agent against UV radiation or sunlight [44,48,49,50,51,53] and in the trial by Fischer et al. [93], who tested melatonin to treat female alopecia. Consequently, further clinical trials using topical melatonin—hopefully based on rigorous experimental settings—are still needed. Future research might take into consideration the use of various concentrations of topical melatonin (without extra active ingredients) as a skin anti-aging agent based on the results of the studies by Sagan et al. [68] and Morganti et al. [69] to obtain a dose–response relationship. Moreover, the work by Fischer et al. [93] (using melatonin to treat female alopecia) suggests that it would be worth extending such investigations to the treatment of male alopecia using the same (or similar) well-designed settings. Future research on topical melatonin might focus on advanced delivery formulations.

## 9. Conclusions

Based on its antioxidant and anti-inflammatory properties, melatonin has been investigated by several research groups for its capability to exert beneficial effects on human skin. These studies were justified for at least three reasons: the skin is characterized by high levels of melatonin and an efficient melatoninergic system; moreover, melatonin can easily penetrate the skin and is practically devoid of allergenic potential. In this present paper, we reviewed the clinical trials in which melatonin was tested alone or combined with additional active ingredients in the same topical formulation as a photoprotective agent, skin anti-aging agent, and hair growth promoter. The studies using melatonin alone offer a clear advantage in terms of information concerning its actual therapeutic efficacy.

The experimental evidence available to date allows us to conclude that melatonin can be safely considered as a valid photoprotective agent with an excellent safety profile. Such a conclusion is based on the results of several clinical trials carried out with a proper design (randomized, double-blind, and placebo-controlled) that employed melatonin alone or in combination with vitamin C and vitamin D. Several studies suggest that melatonin might be endowed with interesting skin anti-aging and hair growth-promoting activities. Unfortunately, most of these studies suffer from limitations in their design (open-label, non-comparative, and melatonin combined with extra ingredients), except for the well-designed work by Fischer et al. [93] in which melatonin was found to be effective in the treatment of some forms of female alopecia. The studies in the above two fields of the application of topical melatonin suggest that further research efforts are needed to draw definitive conclusions concerning the actual benefits of this natural substance to counteract skin aging and hair loss.

## Figures and Tables

**Figure 1 ijms-25-05167-f001:**
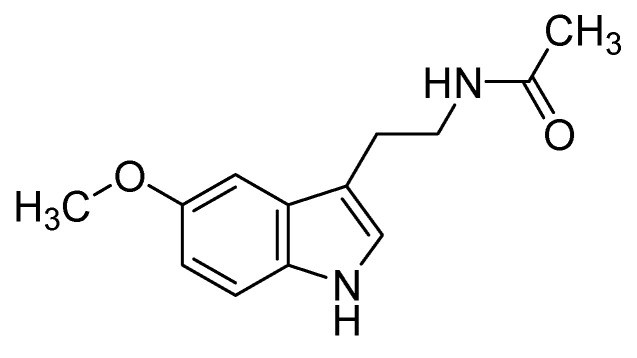
Structure of melatonin.

**Figure 2 ijms-25-05167-f002:**
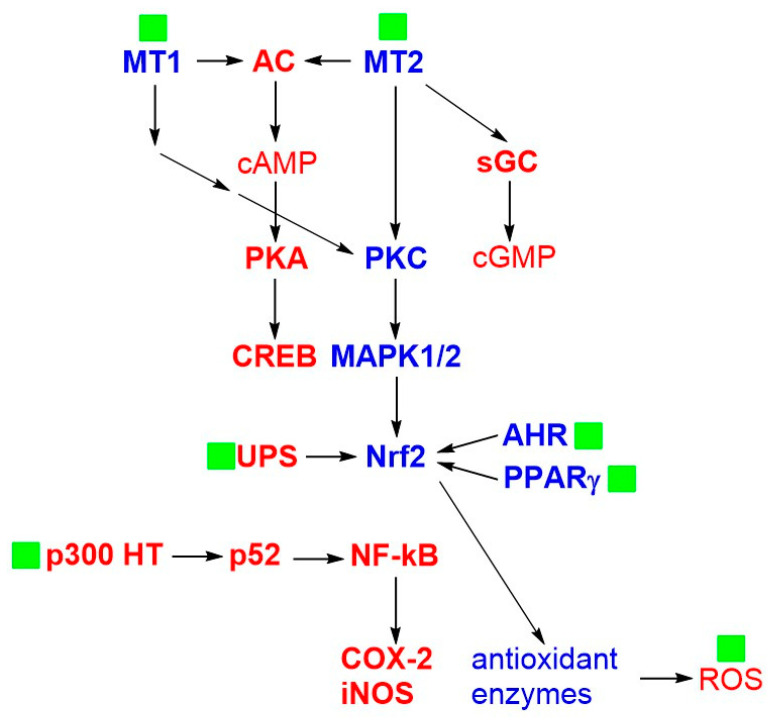
The main mechanisms of action of melatonin underlying its antioxidant and anti-inflammatory properties (see text for details). Melatonin is represented as a green filled square; the proteins that are positively modulated or upregulated are in blue; and the proteins that are negatively modulated or downregulated are in red. The levels of the second messengers and enzymes are in blue or red to indicate, respectively, their increase or decrease. Abbreviations (listed in the same order as they are mentioned in the text): MT1: melatonin receptor 1; MT2: melatonin receptor 2; PKA: protein kinase A; CREB: cAMP responsive element-binding protein; PKC: protein kinase C; MAPK1/2: mitogen-activated protein kinases 1 and 2; Nrf2: nuclear factor erythroid 2-related factor 2; AHR: aryl hydrocarbon receptor; PPARγ: peroxisome proliferator-activated receptor γ; UPS: ubiquitin–proteasome system; ROS: reactive oxygen species; COX-2: cyclooxygenase-2; iNOS: inducible nitric oxide synthase; p300 HAT: p300 histone acetyltransferase; NF-κB: nuclear factor kappa-light-chain-enhancer of activated B cells; AC: adenylate cyclase; and sGC: soluble guanylate cyclase.

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
