# Peer review of "Clinical Studies Using Topical Melatonin"

_ijms, 2024, doi:10.3390/ijms25105167_

Round 1

Reviewer 1 Report

Comments and Suggestions for Authors

This is an important review that would be of strong interest to the readers of the journal.

The paper for most part is well written. However, I suggest some corrections that would improve the manuscript, see below.

The factual error on reporting that RZR/RORA is a receptor for melatonin has to be corrected. Crystal structure of ROR is known for years and it is a receptor for sterols. It can also bind vitamin D metabolites, however, it does not bind melatonin  (FASEB J 28:2775-2789, 2014). The nuclear recepor fro melatonin has been recently identified which includes aryl hydrocarbon receptor (AhR)(doi: 10.3390/ijms242015496). Tghis should be mentioned since photopotective effects of melatonin are independent of MT1 and MT2  (Sci Rep 2017;7(1):1274, 2017).

I suggest to correct the subtitles. For example list different subtitle for Results. This is a review not experimental paper.

Number of references for review is moderate

Author Response

Referee n. 1

This is an important review that would be of strong interest to the readers of the journal.

The paper for most part is well written. However, I suggest some corrections that would improve the manuscript, see below.

The factual error on reporting that RZR/RORA is a receptor for melatonin has to be corrected. Crystal structure of ROR is known for years and it is a receptor for sterols. It can also bind vitamin D metabolites, however, it does not bind melatonin  (FASEB J 28:2775-2789, 2014). The nuclear recepor fro melatonin has been recently identified which includes aryl hydrocarbon receptor (AhR)(doi: 10.3390/ijms242015496). Tghis should be mentioned since photopotective effects of melatonin are independent of MT1 and MT2 (Sci Rep 2017;7(1):1274, 2017).

I suggest to correct the subtitles. For example list different subtitle for Results. This is a review not experimental paper.

Number of references for review is moderate.

1) The factual error on reporting that RZR/RORA is a receptor for melatonin has to be corrected. Crystal structure of ROR is known for years and it is a receptor for sterols. It can also bind vitamin D metabolites, however, it does not bind melatonin  (FASEB J 28:2775-2789, 2014). The nuclear recepor fro melatonin has been recently identified which includes aryl hydrocarbon receptor (AhR)(doi: 10.3390/ijms242015496). Tghis should be mentioned since photopotective effects of melatonin are independent of MT1 and MT2 (Sci Rep 2017;7(1):1274, 2017).

The pharmacological section (3. Pharmacology of melatonin) has been substantially rewritten to take entirely into account the detailed observation of the referee. This section has been considerably improved thanks to his/her suggestions also by adding new references and a new scheme depicted in Figure 2 summarizing the basic pharmacology of melatonin.

2) I suggest to correct the subtitles. For example list different subtitle for Results. This is a review not experimental paper.

Our intention was to entitle the section describing the basic pharmacology of melatonin “Pharmacology of melatonin” rather than “Results” but something went wrong while editing the file. We apologize for this typo. In the revised version of the manuscript this subtitle is written correctly.

3) Number of references for review is moderate.

The number of references has been increased (from 90 to 100). The pharmacological section now reports not only the references suggested by this Referee (see point n. 1) but also several references added to make some points more detailed.

Reviewer 2 Report

Comments and Suggestions for Authors

In the manuscript, consistent with the title and aim, the Authors have compiled information on clinical trials of topical melatonin use.  Since there are few clinical trials, the text of the manuscript is quite simple and is mainly based on a rather detailed description of the presented studies.

I am quite concerned that the manuscript in this form may not meet the requirements of the Editors of the Journal.

Additional comments:

1.      The table needs improved formatting, as it is illegible in this form.

2.      In the text, there is repeatedly a wrong spelling of single words, e.g. in-creases, interleu-kin-1 etc.

Author Response

Referee n. 2

In the manuscript, consistent with the title and aim, the Authors have compiled information on clinical trials of topical melatonin use.  Since there are few clinical trials, the text of the manuscript is quite simple and is mainly based on a rather detailed description of the presented studies.

I am quite concerned that the manuscript in this form may not meet the requirements of the Editors of the Journal.

Additional comments:

1.      The table needs improved formatting, as it is illegible in this form.

2.      In the text, there is repeatedly a wrong spelling of single words, e.g. in-creases, interleu-kin-1 etc.

1) The table needs improved formatting, as it is illegible in this form.

In the current version of the manuscript, the content of Table 1 has been splitted into three different tables and rows are now separated by lines to improve readability. Each of the three tables refers to a specific field of melatonin applications, specifically Table 1 photoprotection, Table 2 sking aging, Table 3 alopecia.

2) In the text, there is repeatedly a wrong spelling of single words, e.g. in-creases, interleu-kin-1 etc.

In the revised version of the manuscript spelling errors have been corrected. 

Reviewer 3 Report

Comments and Suggestions for Authors

This paper is a review of topical applications of melatonin for skin protection, anti-aging, and hair health. It covers an interesting topic and provides useful information about the current research status and application potential of melatonin. However, the composition and quality of the paper need to be improved.

1. Add a separate chapter for Methods on how you selected papers in advance to write a review. Describe the databases and keywords used to search papers, and the conditions for inclusion and exclusion of papers. If possible, draw a flowchart and describe this process in detail.

2. I propose to change the title of the current chapter 2 Results to Mechanistic basis of pharmacological action of melatonin action. Also, please present a picture illustrating the pharmacological action principle of melatonin.

3. In the current chapters 4, 5, and 6, please add a table and a figure to each chapter, to summarize critical article and to schematize key concept. 

4. Please add a new chapter to describe the limitations of existing research and future prospects. Please list and explain in detail the problems that have not been solved/problems that need to be solved in the skin application research of melatonin.

Author Response

Referee n. 3

This paper is a review of topical applications of melatonin for skin protection, anti-aging, and hair health. It covers an interesting topic and provides useful information about the current research status and application potential of melatonin. However, the composition and quality of the paper need to be improved.

1. Add a separate chapter for Methods on how you selected papers in advance to write a review. Describe the databases and keywords used to search papers, and the conditions for inclusion and exclusion of papers. If possible, draw a flowchart and describe this process in detail.

2. I propose to change the title of the current chapter 2 Results to Mechanistic basis of pharmacological action of melatonin action. Also, please present a picture illustrating the pharmacological action principle of melatonin.

3. In the current chapters 4, 5, and 6, please add a table and a figure to each chapter, to summarize critical article and to schematize key concept.

4. Please add a new chapter to describe the limitations of existing research and future prospects. Please list and explain in detail the problems that have not been solved/problems that need to be solved in the skin application research of melatonin.

1) Add a separate chapter for Methods on how you selected papers in advance to write a review. Describe the databases and keywords used to search papers, and the conditions for inclusion and exclusion of papers. If possible, draw a flowchart and describe this process in detail.

In the revised version of the manuscript, we added a chapter entitled “2. Methods” to summarize in detail how we searched the Pubmed database (search query, filters, exclusion criteria) to identify clinical trials relevant to the topic covered by our review.

2) I propose to change the title of the current chapter 2 Results to Mechanistic basis of pharmacological action of melatonin action. Also, please present a picture illustrating the pharmacological action principle of melatonin.

This point has been addressed (see also replay to point n.2 raised by Referee n. 1). The chapter is now entitled “3. Pharmacology of melatonin” and a picture illustrating the molecular mechanisms of action of melatonin has been inserted (Figure 2).

3) In the current chapters 4, 5, and 6, please add a table and a figure to each chapter, to summarize critical article and to schematize key concept.

The chapters to which this Referee refers are now accompanied by a specific table summarizing the critical articles described and discussed. Moreover, as mentioned, a new figure (Figure 2) has been added to schematize key concepts concerning the mechanisms of action of melatonin underlying its use as a photoprotective agent, sking anti-aging agent, and hair growth promoting agent which are the topics of three distinct chapters.

4) Please add a new chapter to describe the limitations of existing research and future prospects. Please list and explain in detail the problems that have not been solved/problems that need to be solved in the skin application research of melatonin.

Throughout the manuscript merits and limitations of the clinical studies have been discussed. However, the point raised by this Referee was a good chance to improve the quality of the manuscript by summarizing - in a new specific chapter - strengths, weaknesses of what has been done in the field of topical melatonin and to provide our view on what could hopefully be done. The new chapter is entitled “Achievements, limitations, and future prospects of clinical studies on topical melatonin”. 

Round 2

Reviewer 2 Report

Comments and Suggestions for Authors

I recommend the manuscript in its current version for publication

Author Response

Referee 2

Thank you very much for your thorough review and for recommending our manuscript for publication. Your support and feedback are greatly appreciated, and we are grateful for the time and effort you've dedicated to evaluating our work.

Warm regards.

Reviewer 3 Report

Comments and Suggestions for Authors

The manuscript has been significantly improved. I suggest some additional revisions before recommending the paper for acceptance. 1. Figure 2. Please remove red and blue arrows as they are unnecessary. Please remove an arrow between melanotin (green square) and ROS for consistency. (Alternatively, you can add arrows between melatonin and its targets). In my opinion, it would be better to indicate melatonin directly rather than using a green square. "negatively modulated or upregulated" should be corrected to "negatively modulated or downregulated". Please add AC, sGC, NF-kB, MT1 and MT2 receptors to the abbreviation list. 2. Table 1, 2, and 3. The column entitled "N" is unnecessary. “Type of outcome” and “statistical significance” columns are not very informative. Please replace them with or add a new column for "measured parameters and differences from the control (p values)". 3. Table 2. Please consider separating “skin aging” into “photoaging”, “natural skin aging”, “intrinsic skin aging”, or “extrinsic skin aging”. 4. Chapter 5 title. "Melatonin as a photoprotective agent" --> "Melatonin as a photoprotective and antiinflammatory agent" 5. Chapter 7 title. "Melatonin to treat" --> "Melatonin as a topical agent to treat".

Author Response

Referee n. 3

We thank this referee for his/her valuable suggestions that allowed us to improve the quality of our manuscript.

The manuscript has been significantly improved. I suggest some additional revisions before recommending the paper for acceptance.

  1. Figure 2. Please remove red and blue arrows as they are unnecessary. Please remove an arrow between melanotin (green square) and ROS for consistency. (Alternatively, you can add arrows between melatonin and its targets). In my opinion, it would be better to indicate melatonin directly rather than using a green square. "negatively modulated or upregulated" should be corrected to "negatively modulated or downregulated". Please add AC, sGC, NF-kB, MT1 and MT2 receptors to the abbreviation list.

Figure 2  has been modified according to the suggestions given by the referee.

  1. Table 1, 2, and 3. The column entitled "N" is unnecessary. “Type of outcome” and “statistical significance” columns are not very informative. Please replace them with or add a new column for "measured parameters and differences from the control (p values)".

Table 1-3 have been modified according to the suggestion given by the referee.

  1. Table 2. Please consider separating “skin aging” into “photoaging”, “natural skin aging”, “intrinsic skin aging”, or “extrinsic skin aging”.

We agree with the referee regarding the different impact of the intrinsic an extrinsic factors on skin aging. Certainly, it would be desirable to study these two factors separately. As stated in chapters 8 of our manuscript, photoaging represents an almost unavoidable extrinsic factor contributing to skin aging up to 80%. Therefore, it is very difficult to classify the examined articles into two well-distinct categories based on intrinsic or extrinsic type of skin aging.

  1. Chapter 5 title. "Melatonin as a photoprotective agent" --> "Melatonin as a photoprotective and antiinflammatory agent"

The title of Chapter 5 has been changed according to the suggestion given by the referee. The title of Table 1 has been has been likewise modified to include the “anti-inflammatory efficacy”.

  1. Chapter 7 title. "Melatonin to treat" --> "Melatonin as a topical agent to treat".

The title of Chapter 7 has been modified according to the suggestion given by the referee.
